# Flexible Electrode by Hydrographic Printing for Surface Electromyography Monitoring

**DOI:** 10.3390/ma13102339

**Published:** 2020-05-19

**Authors:** Xiong Zeng, Ying Dong, Xiaohao Wang

**Affiliations:** 1Tsinghua Shenzhen International Graduate School, Tsinghua University, Shenzhen 518055, China; zengx17@mails.tsinghua.edu.cn (X.Z.); wang.xiaohao@sz.tsinghua.edu.cn (X.W.); 2Tsinghua-Berkeley Shenzhen Institute, Tsinghua University, Shenzhen 518055, China

**Keywords:** dry electrode, flexible device, fractal pattern, hydrographic printing, sEMG

## Abstract

Surface electromyography (sEMG) monitoring has recently inspired new applications in the field of patient diagnose, rehabilitation therapy, man–machine–interface and prosthesis control. However, conventional wet electrodes for sEMG recording cannot fully satisfy the requirements of these applications because they are based on rigid metals and conductive gels that cause signal quality attenuation, motion artifact and skin allergy. In this study, a novel flexible dry electrode is presented for sEMG monitoring. The electrode is fabricated by screen-printing a silver–eutectic gallium–indium system over a transfer tattoo paper, which is then hydrographically printed on 3D surface or human skin. Peano curve in open-network pattern is adopted to enhance the mechanics of the electrode. Hydrographic printing enables the electrode to attach to skin intimately and conformably, meanwhile assures better mechanical and electrical properties and therefore improves the signal quality and long-term wearability of the electrode. By recording sEMG signal of biceps under three kinds of movement with comparison to conventional wet electrode, the feasibility of the presented flexible dry electrode for sEMG monitoring was proved.

## 1. Introduction

Flexible devices or epidermal electronics are newly developed wearable electronics, which are much more lightweight, robust and flexible in contrast to traditional devices, therefore allowing them intimately mounted on human skin [1,2,3,4,5]. These skin-like sensing systems provide opportunities for long-term and continuous measurement of bioelectrical signals such as electrocardiogram (ECG), electroophthalmic (EOG), electromyography (EMG) and physical parameters like temperature and humidity [5,6,7,8,9].

EMG signal is generated by the electrical activity of the muscle fibers that are activated almost synchronously by the motor neuron innervating them [10,11,12]. EMG can be detected by intramuscular electrodes [13], known as needle EMG or by electrodes mounted on skin [14], known as surface Electromyography (sEMG). As one of the most common signals non-invasive, sEMG is widely used to not only detect neurophysiology and analyze movement or gait, but also interface with man-machine and control prosthesis [8,15,16,17].

The most common and commercially available electrodes for bioelectrical signals acquisition are wet conductive gel electrodes. While they are characterized by causing irritation and allergy, signal degradation during long-term monitoring, as well as difficulty in adhering skin without adhesive. Dry flexible electrode is a promising alternative as they can attach to skin with intimate attachment, less motion artifact and no skin reaction [18]. While most dry flexible electrodes are typically based on polymer [19] (PDMS [20], PI [21]), metal (Au [22], AgNP [23]), carbon [24,25] and liquid metal (AlGaN/GaN [26], EGaIn [27,28,29]). Previous works also typically proved that rigid metal materials (Au [30], Cu [31]) with serpentine and bridge structure [32] could better tolerate mechanical deformations such as stretching, compressing and twisting. These materials are usually fabricated through lithography technology, chemical vapor deposition (CVD), physical vapor deposition (PVD) and other patterning techniques that are expensive and require complex processes and high-quality conditions [33,34,35,36]. While hydrographic printing technology transferring the pattern and material on the medium through water is uncomplicated and inexpensive, which has been used to process flexible electrodes on skin [37,38,39,40,41].

In this work, a novel flexible dry electrode for sEMG monitoring is presented. In the following sections, the pattern, structure, fabrication process and characterization of the electrode are introduced. Then the contact impedance and the stretch–resistance characteristics of the electrode are tested. Finally, the feasibility of the electrode for sEMG monitoring is verified through recording sEMG signal of biceps under three kinds of movement and testing the long term wearability in comparison to traditional wet electrode.

## 2. Pattern and Structure

Considering the generation and location of sEMG, the electrode pattern and size are well designed according to the guideline of SENIAM (surface electromyography for the non-invasive assessment of muscles) project. The electrode pattern consists of three phases, i.e., reference (REF), ground (GND), record (REC) phase, as shown in Figure 1a, from left to right. To assure that the electrode remains functional under strains of up to 30% which is the typical stretch ability of natural human skin and high signal noise ratio, each phase shapes in the first-order Peano curve [42], with the total dimensions of 5 mm × 5 mm and wire width of 100 μm. The end pad of each phase, with dimension of 1 mm × 2 mm, provides an electrical interface for data acquisition through ACF connector (M-505A, JieMai precision automation co. LTD, ShenZhen, China). The electrode pattern is printed on flexible tattoo paper (silhouette temporary tattoo paper, American, INC, Lindon, UT, USA) that is composed of an adhesive sheet and a support sheet. The adhesive sheet contains slip layer, glue layer and support paper layer, as shown in Figure 1b. The glue layer is able to assure the electrodes intimately attaching to skin. While used as the loading carrier of the electrodes, the support sheet usually includes ethyl cellulose (EC) layer, soluble layer and support paper layer, as shown in Figure 1c. There will only be the EC layer support the pattern and material left after dissolution. With the employment of the paper material, the pattern and structure have the ability to intimately attach to skin surface without damage, meanwhile block the oxygen influencing the sensitive complex system.

## 3. Processes and Characterization

The electrode is manufactured by hydrographic printing silver–indium–gallium alloy system (silver, ET-5A, UVTM, 3.9 GPa; indium–gallium, 24%–76%) on the tattoo paper. It has been found out that ternary alloy system silver–indium–gallium would form when silver is added into eutectic Ga–In at atmosphere condition [43]. Weak acid can be used to clean the alloy system if there are extra materials. As illustrated in Figure 2, the hydrographic printing process is composed of six steps. First, the adhesive sheet was laser (current 5 A, velocity 5 m/s) patterned and the silver–indium–gallium alloy system was screen-printing patterned on the support sheet. After the alloy system solidified, the adhesive sheet and the support sheet were bonded by the glue layer. Then the bonding system was soaked in water and after 30 s, the water-soluble middle layer of the tattoo paper substrate dissolved and separated the flexible film from the support paper. The left flexible film then attached to a 3D object or skin, but the surface of the flexible film was still moist. In order to reduce the oxidation of the alloy system due to the moist, it is better to take about 5 min to get the wet surface dry out in ambience. Moreover, the time can be shortened if heater is used.

After finishing the electrode, scanning electronic microscopy (SEM) and energy dispersive X-ray spectroscopy (XPS) were taken to examine the surface and material characteristics of the traces. As shown in Figure 3b, the thickness of the silver–indium–gallium trace only adds by 17 μm in contrast to only silver paste system, but the silver–indium–gallium trace has more pores after treated with the weak acid solution, as shown in Figure 3a. Hence, it’s better to guarantee the cleaning process fulfilling in a very short time. The XPS survey spectra is shown in Figure 4a. The spectrum shows gallium, silver and indium peaks [44,45], which are zoomed in and shown in Figure 4b–d, respectively. For Ga, peaks are observed at ~14.3 ev and 20.1 ev, which associates with silver–indium–gallium system and oxidation. For Ag, peaks are observed at ~362.8 ev and 368.4 ev, which associates with silver–indium–gallium system and oxidation. For In, peak is observed at ~451.8 ev, which associates with silver indium–gallium system. These results indicate that the system will lead to oxidation that can lower down the conductivity. Weak acid solution usually is applied to eliminate the influence [38].

## 4. Test results and Discussion

### 4.1. Electrical and Mechanical Performances

The ability of electrodes to record sEMG depends strongly on the skin–electrode contact impedance. Comparative experiments between traditional wet electrode and the flexible dry electrode are carried out, which demonstrates that the contact impedance declines with the increasing intimacy. Figure 5 shows the practical application scenarios, the skin contact models and the equivalent circuits of the electrodes. The relative impedance data can be analyzed with the equivalent circuit that consists of electrode’s resistor and coupling circuit. As illustrated in Figure 6, the tested contact impedance between skin and the flexible dry electrode is lower than that of the traditional wet electrode under 1000 Hz in which sEMG signal mainly distributes. Moreover, the impedance of the flexible electrode is much lower than that of the traditional electrode under 500 Hz, which is the central power range of sEMG. The result proves that the flexible electrode is mounted much more intimately on skin.

To better understand the influence of the mechanical deformation on electrical characteristics, the resistance of an eight-millimeter trace under stretch was examined. Referring to Figure 7a–c, the measured resistance shows a different performance in the different strain ratio. The lower strain ratio presents the better repeatability. However, the resistance always keeps at extremely low level. For each loading cycle, the peak resistance increases with the increased displacement, but produced regression and sharp dropping under 30% strain ratio experiments. These unexpected circumstances are mainly due to the postponement of the measurement, inertia existing in the trace under relaxing and irreversible deformation under great strain ratio. Overall, according to the relative resistance change (Figure 7d) as a function of strain, the trace still maintains excellent electrical characteristics without enormous variation even though the exact resistance has a slight rising. In contrast, this flexible electrode enables intimately and conformably mounting to skin with good electromechanical characteristics.

### 4.2. sEMG Monitoring Tests

The electrical and mechanical features are summarized above as the fundamental for the sEMG signal measuring. The primary application test is to measure sEMG signal of biceps under voluntary contraction [46]. Three contrast experiments were taken to justify the practical signal quality of the flexible electrode. The Biopac MP36 sEMG scope was used to record the sEMG signal and two channels were separately connected to the traditional wet electrode and the flexible dry electrode at the same time. First, the electrodes measure 5 fist moves with the proper interval and the results are shown in Figure 8a and Appendix A. Second, the movements are continuous fists without intervals and the results were shown in Figure 8b and Appendix A. Third, the movements include not only the fist, but also the arm flex and the results were shown in Figure 8c and Appendix A. It shows that the flexible electrodes captured the signals of the muscle fiber precisely. However, the noise of the baseline is higher than the commercial traditional electrodes due to the unshielded connector between the flexible electrode leading wire and the Biopac MP36 sEMG scope. Nevertheless, the flexible electrodes are enough suitable to sEMG signal recording.

### 4.3. Wearability

In addition to mechanical and electrical property tests, the conformable long-term wearability also takes a critical role in sEMG monitoring. Figure 9 shows the comparison of the wearable effects between the flexible and traditional electrodes. The skin zones of the traditional electrodes appear redness and swelling after the sEMG monitoring experiments. However, the more comfortable flexible electrodes only leave the shadow of the excess liquid metal.

## 5. Conclusions

An ultrathin, flexible and skin conformable dry electrode that can be applied without skin preparation was presented. The electrode is fabricated through hydrographic printing which can magnificently lower the cost and simplify the processing. Due to the novel structure and elaborate pattern, the electrode can susceptibly attach to the skin like a tattoo. This unique feature assures better signal quality and long-term wearability of the electrode in sEMG recording. In the comparative tests, the flexible dry electrode performs equally well or even better than traditional wet electrode, especially on aspects of skin–electrode contact impedance and wearability. Even though the ACF connector used in the experiments introduces a baseline noise, the dry electrode performs well under movement. In conclusion, the flexible dry electrode can be expected for sEMG monitoring applications.

## Figures and Tables

**Figure 1 materials-13-02339-f001:**
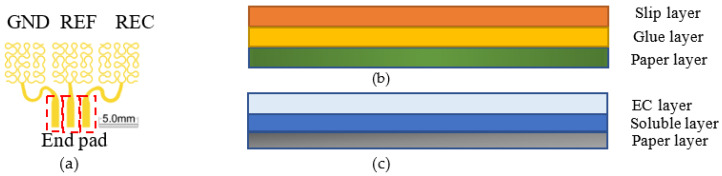
The fractal pattern of the flexible electrodes and the transfer tattoo paper. (**a**) overall pattern of the first-order Peano curve. (**b**) the adhesive sheet and (**c**) support sheet of the transfer tattoo paper.

**Figure 2 materials-13-02339-f002:**
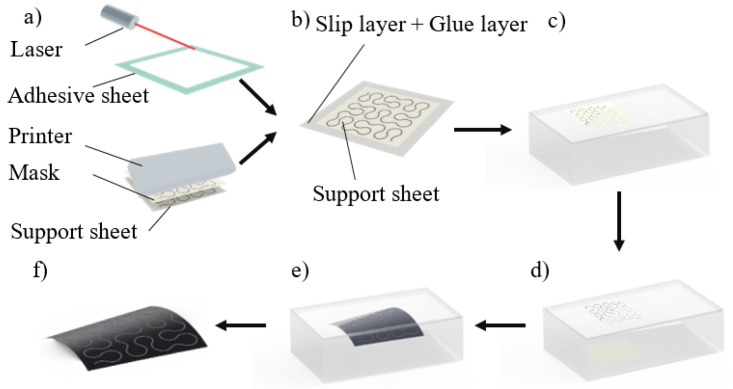
Fabrication processes of hydrographic printing. (**a**) materials are processed by laser pattern and screen-printing; (**b**) adhesive is attached to the screen-printing pattern; (**c**) complex is soaked in water; (**d**) paper layer of the support detaches from the complex; (**e**) left flexible film is transferred to a 3D object or skin; (**f**) finished electrode.

**Figure 3 materials-13-02339-f003:**
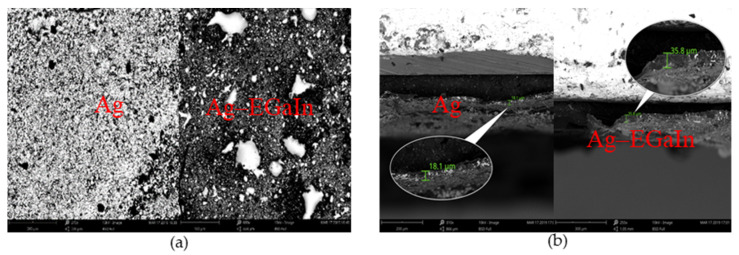
Characteristics of scanning electronic microscopy (SEM) (**a**) left is the only silver trace surface and the right is the Ag–EGaIn trace surface (**b**) right is the silver trace cross section and the left is the Ag–EGaIn trace cross section.

**Figure 4 materials-13-02339-f004:**
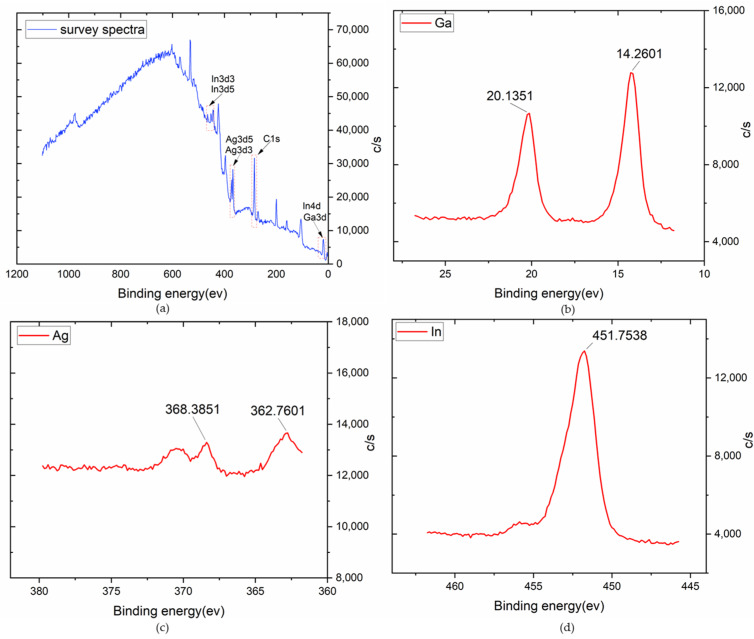
Characteristics of energy dispersive X-ray spectroscopy (XPS) (**a**) survey spectra; (**b**) binding energy of gallium; (**c**) the binding energy of silver (**d**) the binding energy of indium.

**Figure 5 materials-13-02339-f005:**
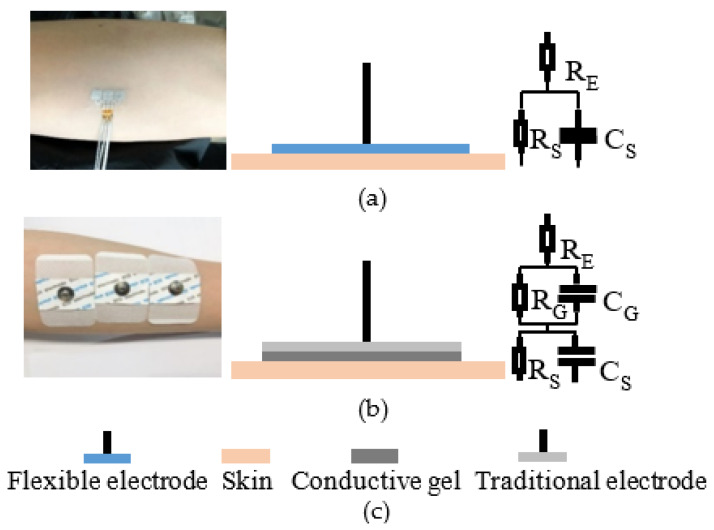
Comparative experiments between the traditional electrode and the flexible electrode. (R_E_: resistance of electrode, R_G_: resistance of gel, C_G_: couple of gel, R_S_: resistance of skin, C_S:_ couple of skin); (**a**) practical application scenario (left), the skin contact model (middle) and the equivalent circuit (right) of the flexible electrodes; (**b**) practical application scenario (left), the skin contact model (middle) and the equivalent circuit (right) of the traditional electrode; (**c**) illustration of the above.

**Figure 6 materials-13-02339-f006:**
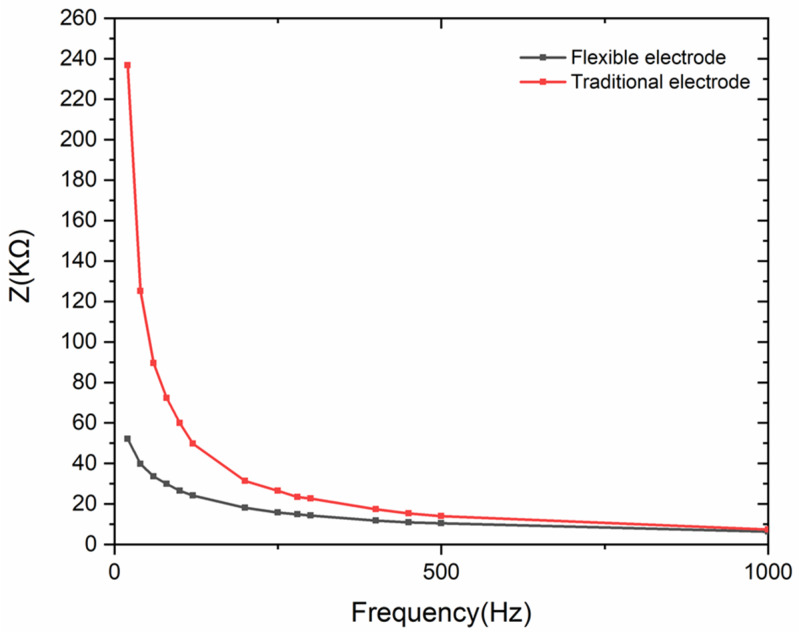
Tested contact impedance between skin and the electrodes at frequency range of 0–1000 Hz.

**Figure 7 materials-13-02339-f007:**
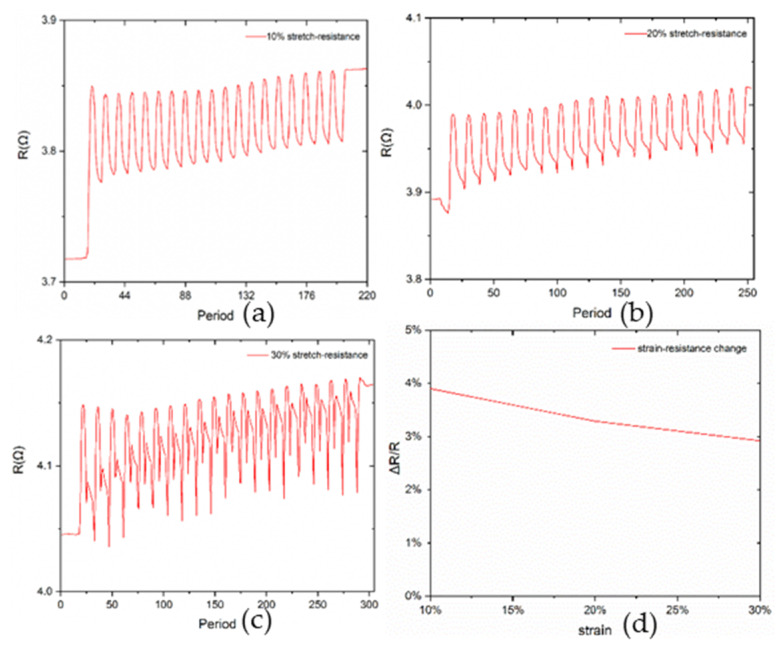
Stretch–resistance test under 10%, 20% and 30% strain for 10 cycles and the relationship between strain and resistance change. (**a**) stretch–resistance test under 10% strain; (**b**) stretch–resistance test under 20% strain; (**c**) stretch–resistance test under 30% strain; (**d**) strain-resistance change relationship of the system.

**Figure 8 materials-13-02339-f008:**
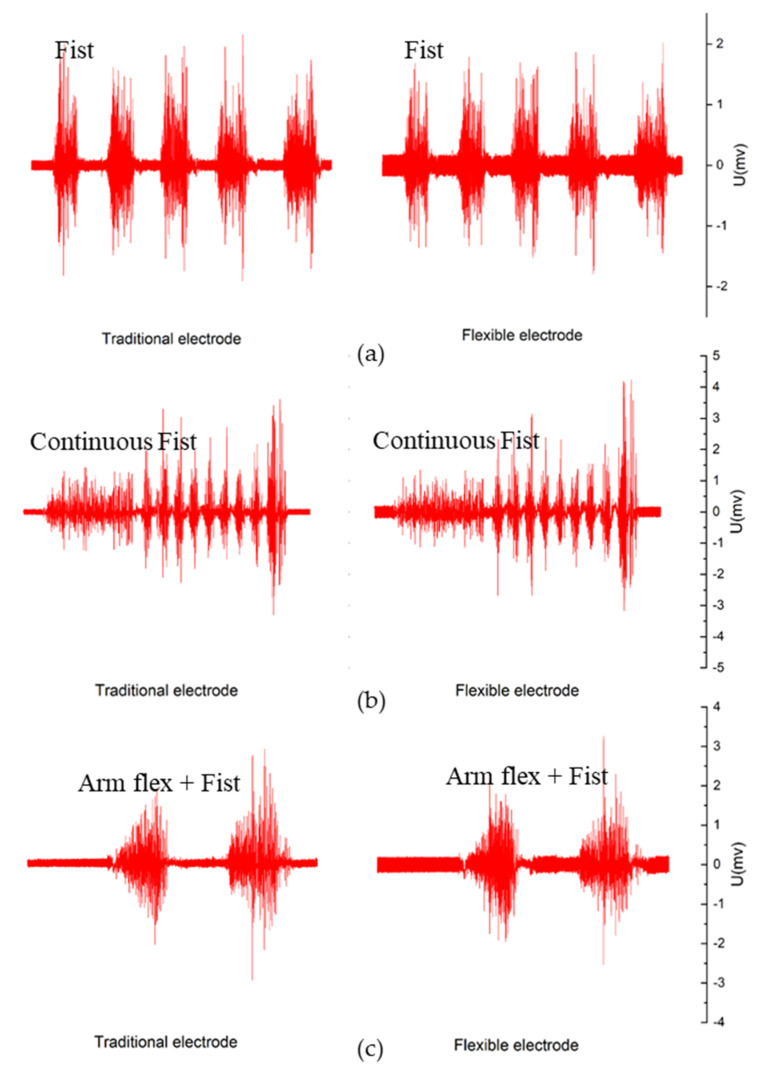
Comparative experiment of sEMG measurements by the traditional wet electrode (left) and the flexible dry electrode (right) (**a**) 5 fist moves with 2-s interval; (**b**) continuous fist moves without interval (**c**) arm flex and first at the same time.

**Figure 9 materials-13-02339-f009:**
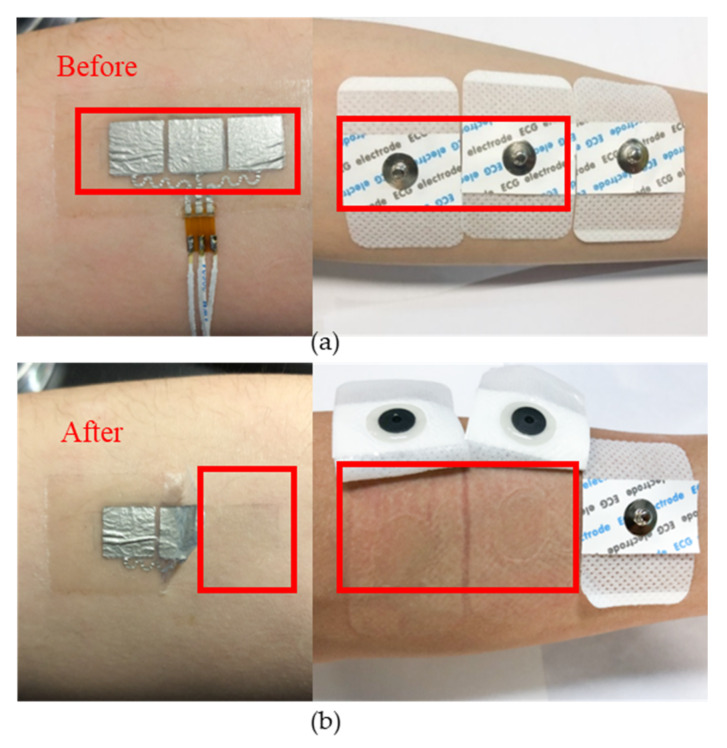
Skin reaction of traditional (right) and flexible (left) electrodes after sEMG measurements (**a**) presentation of the flexible electrodes and the traditional electrodes before the experiments; (**b**) presentation of the skin contact areas after the experiments.

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
