# Peer review of "Flexible Electrode by Hydrographic Printing for Surface Electromyography Monitoring"

_materials, 2020, doi:10.3390/ma13102339_

Round 1

Reviewer 1 Report

Dear Authors,

In the reviewed attachment I suggeste to correct some misprint

and to provide more info in a figure of the paper.

All the othe parts are well described and clearly presented.

My best wishes

Author Response

Thank you very much for the valuable suggestions and kind comments. All the mentioned misprints have been corrected, and the zoom on y axe of Figure 7b has been improved. We appreciate the suggestion “Possibly show curves at 20% -30% adding to 7a”, and the curves have been supplemented in the revision.

Reviewer 2 Report

The manuscript presents a low cost electrode for Surface Electromyography (sEMG) recording. The manufacturing method is the highlight of the manuscript. However, it is not clearly explained. Some of the figures labels (figure 2) are cropped, nonstandard terminology is used, there is a lot of grammatical errors and inaccurate use of the English words and phrase. The English needs a lot of improvement. The manuscript is premature up to a level that makes it very hard to review. Abstract is written more like as introduction, the results have not been highlighted. In abstract states that the current electrodes results in better signal to noise ratio but in Page 6, it is indicated that the signal to noise ratio of the presented electrode is lower than conventional electrodes. The spacing between working and reference electrodes is very small while this spacing generally is at least 1.5 cm. Overall, I believe the manuscript is not ready for publication.

Author Response

We appreciate your valuable suggestions and have carefully modified the manuscript according to your comments. Please allow us to break the comments down into four questions and response them one by one.

Q1, The manufacturing method is the highlight of the manuscript. However, it is not clearly explained.

A: The first paragraph in section 3 has been rewritten to introduce the manufacturing method in detail.

Q2, Some of the figure labels (figure 2) are cropped, nonstandard terminology is used, there is a lot of grammatical errors and inaccurate use of the English words and phrase. The English needs a lot of improvement. The manuscript is premature up to a level that makes it very hard to review. Abstract is written more like as introduction, the results have not been highlighted.

A: The manuscript has been carefully checked and modified. Errors in figure labels, terminologies, grammars, words and phrases have been corrected. Abstract and conclusion have been rewritten.

Q3, In abstract states that the current electrodes result in better signal to noise ratio but in Page 6, it is indicated that the signal to noise ratio of the presented electrode is lower than conventional electrodes.

A: We apologize for the misleading expression and thank you very much for pointing out this mistake. The presented electrode does show better properties than conventional electrodes in the respect of skin contact impedance and wearability. However, a baseline noise due to the ACF connector can be observed in the experimental results, which failed to prove the conclusion “better signal to noise ratio”. The expression of the results has been modified.

Q4, The spacing between working and reference electrodes is very small while this spacing generally is at least 1.5 cm.

A: We appreciate your professional comment. In this work the spacing was set according to the references ([14], SENIAM project), and since acceptable results were achieved, we did not take this point into consideration. Thank you for the valuable comment and we will do research on this point when we can go back to the lab (We’ve not been permitted to go back to the university yet because of the epidemic).

Reviewer 3 Report

Q1.) In page 2, It is better if the author explains the full terms first (CVD, PVD) then writes the term in abbreviation. 

Q2.) XPS survey spectra should be added

Q3.) XPS-High resolution spectra of silver trace surface and the Ag-EGaIn trace surface needs more explanation.

Q4.) It is better to add the EDX analysis of the silver trace surface and the Ag-EGaIn trace surface that further confirm the trace elements.

Q5.) Add high resolution SEM images as inset in figure.3 for better morphology analysis.

Q6.) Add the phase (Ф) Of the test electrodes on the skin in the inset of the figure. 6.

Q7.) Increase the test cycles for 40 cycles and strain >10% and check the relationship between strain and the resistance change.

Q8.) The signals coming from the traditional electrode and the flexible electrodes in figure 8. are similar, better show the three (n=3) independent measurement graphs for fist, continuous fist and Arm flex+fist sEMG.

Q9.) English should be polished more. Proof by the special language coordinator is required.

Q10.) Rewrite the conclusions by adding the limitation of the study.

Author Response

Thank you very much for the professional comments and valuable suggestions. We have tried our best to supplement information and modify the paper. However, some of the suggestions can not be implemented right now because we can not go back to the university to do the experiments because of the epidemic. Here please allow use to response the questions one by one.

A1.) The full terms have been explained where they appeared for the first time.

A2.) XPS survey has been added into Figure 4.

A3.) More explanations about the XPS survey result has been added in section 3.

A4.) Thank you very much for this valuable suggestion. We are very sorry that we can not do the EDX analysis right now because we have not been permitted to go back to the university yet due to the epidemic situation.

A5.) This is another valuable suggestion that we can not implement right now. But we will do more research on this point when we can go back to the lab.

A6.) We are very sorry that we did not take the phase of the impedance into consideration and did not record the data. But we believe this can provide more information about the electrical characteristics of the electrode. We will do the experiment again as we can go back to the lab.

A7.) Curves at 20% strain and 30% strain have been added to Figure7 in the revision, and the relationship between strain and the resistance change has been analyzed. We will do more experiment in this point with increased test cycles when we can go back to the lab.

A8.) Thank you for this valuable suggestion. Due to the page limitation, we have provided the three independent graphs for each movement as supplement of the revised manuscript.

A9.) The manuscript has been carefully checked and modified. We have tried our best to polish the language and hope the revision is acceptable.

A10.) We appreciate this suggestion and have rewritten the conclusion to explain the limitations of the study.

Reviewer 4 Report

The manuscript describes a new and interesting methodology for the fabrication of wearable electrodes based on hydrographic printing to be used in Surface Electromyography monitoring. The new electrodes show comparable performance as that of conventional gel devices with improved flexibility, impedance and respect for the skin confort. This is why I consider that the manuscript can be accepted for publication practically as it is, with the exception of a few typografic corrections:

  • In the second paragraph of the introduction 'without invasive' should be replaced by 'non-invasive'.
  • In the caption of Figure 8 there are some typing issues: 'elelctrod' instead of 'electrode', 'traditiaon' instead of 'traditional' and 'first' instead of 'fist'.

Author Response

Thank you very much for the valuable suggestions and kind comments. All the mentioned misprints have been corrected and the manuscript has been carefully checked and modified.

Round 2

Reviewer 3 Report

The authors revised the manuscript accordingly to the reviewers suggestions and presented the paper nicely.